# Feasibility of Using a MEMS Microphone Array for Pedestrian Detection in an Autonomous Emergency Braking System

**DOI:** 10.3390/s21124162

**Published:** 2021-06-17

**Authors:** Alberto Izquierdo, Lara del Val, Juan J. Villacorta

**Affiliations:** 1Department of Signal Theory and Communications and Telematics Engineering, University of Valladolid, Paseo de Belén, 15, 47011 Valladolid, Spain; juavil@tel.uva.es; 2School of Industrial Engineering, University of Valladolid, Paseo del Cauce, 59, 47011 Valladolid, Spain

**Keywords:** microphone array, pedestrian detection, MEMS, autonomous emergency braking system

## Abstract

Pedestrian detection by a car is typically performed using camera, LIDAR, or RADAR-based systems. The first two systems, based on the propagation of light, do not work in foggy or poor visibility environments, and the latter are expensive and the probability associated with their ability to detect people is low. It is necessary to develop systems that are not based on light propagation, with reduced cost and with a high detection probability for pedestrians. This work presents a new sensor that satisfies these three requirements. An active sound system, with a sensor based on a 2D array of MEMS microphones, working in the 14 kHz to 21 kHz band, has been developed. The architecture of the system is based on an FPGA and a multicore processor that allow the system to operate in real time. The algorithms developed are based on a beamformer, range and lane filters, and a CFAR (Constant False Alarm Rate) detector. In this work, tests have been carried out with different people and in different ranges, calculating, in each case and globally, the Detection Probability and the False Alarm Probability of the system. The results obtained verify that the developed system allows the detection and estimation of the position of pedestrians, ensuring that a vehicle travelling at up to 50 km/h can stop and avoid a collision.

## 1. Introduction

Nowadays, road vehicles are the most widespread transportation system. Current estimations set the mortality due to traffic accidents and collisions at around 1.35 million people per year. Furthermore, 23% of this mortality is associated with pedestrian fatalities, totaling some 310,000 pedestrian deaths per year [1]. To try to reduce this fatal number, the automotive and transport sectors are actively working on different solutions, even more so since the development of autonomous cars began, where their use in urban environments is still an unsolved problem due to the high presence of pedestrians.

On the one hand, among the different actions to reduce the number of pedestrian accidents that are being carried out, some of these actions focus on studies related to the analysis of pedestrian behavior at crosswalks [2,3,4,5], to try to anticipate their movements, and take them into account when making decisions about braking the vehicle or slowing down its speed in areas likely to be at risk, due to the presence of pedestrians, who could become potential victims. This information is very valuable in terms of paying attention to pedestrian detection and tracking [4] and also in terms of designing routes to avoid potential risk areas as much as possible [5].

On the other hand, further actions to reduce the number of pedestrian accidents are focused on the development and improvement of intelligent transport systems. Many of these systems pay special attention to the interaction between vehicles and pedestrians, warning vehicle drivers that pedestrians are in the vicinity [6,7,8]; other systems are centered on information sharing between vehicles and the transport infrastructure itself [9,10].

In addition to these actions, over the last several decades, vehicles have been equipped with Advanced Driver Assistance Systems (ADAS), such as ABS (Anti-Blocking System), ESC (Electronic Stability Controllers), and AEB (Autonomous Emergency Braking), which improve their comfort, efficiency, and safety. All these systems are of particular interest in autonomous vehicles. Among these systems, there are AEB systems specifically focused on pedestrian detection, the so-called AEB-P (AEB for Pedestrians), whose studies are heavily focused on trying to reduce the number of accidents involving pedestrians, either by using monocular cameras [11], analyzing time-to-collision (TTC) [12,13], mapping the positions of detected pedestrians [14], including vacuum emergency braking (VEB) [15], or developing protocols for testing emergency braking systems [16,17].

For all these actions mentioned above, studies and developments are essential to improve pedestrian detection systems and algorithms, either by using statistical models [18], combining the use of different detection systems [19], defining complex scenarios [20], improving algorithms for tracking detected pedestrians [21,22], developing algorithms for detecting and tracking multiple pedestrians [23,24,25], or by using machine learning algorithms [26,27,28].

Most of the systems used for pedestrian detection are based on RGB cameras and image processing algorithms. These systems are very effective when the surrounding visibility conditions are adequate, but when the visibility is reduced, their performance drops considerably. For this reason, several studies are currently focused on trying to solve this problem. Some of the solutions being worked on are based on improving pedestrian detection algorithms in this type of low visibility environment [29,30,31], and others focus on using other types of systems to obtain information, such as the use of LIDAR [32,33] or infrared sensors [34,35], or on fusing images obtained from the classic RGB camera with other detection systems, such as thermal cameras [36], LIDAR [37], or an array of microphones [38].

Under these premises, the idea of analyzing the feasibility of detecting pedestrians by means of an acoustic array installed in a vehicle arose. The system, working in conjunction with AEB systems, would prevent traffic accidents. Using acoustic signals ensures that the system will function properly in environments with reduced visibility. Arrays are ordered sets of identical sensors, whose response pattern is controlled by modifying the amplitude and phase given to each sensor, their spacing, and their distribution in space (linear, planar, etc.) [39]. Beamforming techniques are used with arrays [40] so that the array pattern is electronically steered to different spatial positions.

The acronym MEMS (Micro-Electro-Mechanical System) refers to technology that develops mechanical systems with a dimension of less than 1 mm on integrated circuits (ICs) [41]. MEMS technology applied to acoustic sensors has developed high-quality microphones with high SNR (signal-to-noise ratio), high sensitivity, and low power consumption [42]. In sound source localization systems, MEMS microphones are often combined with FPGA-based architectures [43].

In this paper, based on the idea that the use of MEMS sensors is common in vehicles and the transport sector [44,45], and on the authors’ previous experience in the implementation of acoustic arrays for human detection [46,47,48], the authors study the feasibility of using an array of MEMS microphones embedded on a vehicle to detect pedestrians. Some tests have been carried out in order to detect the position of pedestrians in outdoor environments. The tests have been carried out at different distances, depending on the braking distances to be considered according to different speeds that a vehicle can carry on urban roads, and also according to the maximum deceleration that the AEB system gives to the vehicle. The main objective of this paper is to analyze the feasibility of using a system with the indicated characteristics to accurately detect pedestrians on a road.

Section 2 introduces the description of the system developed in this study, showing the different hardware platforms that compose the system and the system features. Section 3 presents the results obtained on the tests of the system and the corresponding discussion of these results. Finally, Section 4 contains the conclusions that authors have drawn on the basis of the obtained results.

## 2. Materials and Methods

This section presents the requirements to be met by the acoustic acquisition and processing system, based on a 2D array of MEMS microphones. Then the hardware on which the system is based is presented, as well as the processing algorithms with which the pedestrian presence is detected.

### 2.1. Requirements

The main objective of the designed system is to allow pedestrian detection using an acoustic array of digital MEMS microphones that will be embedded on a vehicle, in order to provide sensing information to an AEB system. To achieve this objective, a set of requirements has been defined that must be satisfied by the system:The designed system will have to be onboard a vehicle, specifically at its front, in order to be able to detect pedestrians in its path.The system will be based on the principle of RADAR (RAdio Detection and Ranging) but using sound waves instead of radio waves, i.e., the system will be based on a SODAR (SOund Detection and Ranging) system. This type of system is an active one, based on the generation of an acoustic signal. This signal is reflected on a pedestrian that could be in the vehicle’s path. The reflected signal is then received by the MEMS microphone array, for its subsequent analysis and processing.The acoustic pedestrian detection system shall be able to communicate with the vehicle’s central vehicle control system to alert the AEB system, so that the car can act accordingly. The reason of this requirement is that the purpose of this system is to be integrated into a vehicle, together with other ADAS systems with which the vehicle may be equipped, to assist the AEB system in making a more reliable braking decision in a pedestrian detection in the vehicle’s path.The system should provide reliable results taking into account that the vehicle may travel at different speeds. In this respect, since AEB systems are often used in urban environments, it has been defined that the system should be able to detect pedestrians for vehicle speeds below 50 km/h, which is the usual speed limit in urban environments (although in some cities, this limit is already being reduced to 30 km/h).Another requirement that the system must comply with is that it must have a good resolution in the horizontal coordinate (azimuth). We assume that the pedestrian will be standing on the road, at the vertical height of the vehicle, so the resolution of the system should be focused on the horizontal coordinate. So, the position of the pedestrian will be represented in this azimuth coordinate.

### 2.2. Hardware Setup

#### 2.2.1. MEMS Array

The acoustic images acquisition system used in this paper is based on a Uniform Planar Array (UPA) of MEMS microphones. This array, which has been entirely developed by the authors and is shown in Figure 1, is a rectangular array, of 5 × 30 MEMS microphones, consisting of 6 array modules. Each of these modules is based on a square array of 5 × 5 sensors, which are uniformly spaced every 0.9 cm in a rectangular Printed Circuit Board (PCB).

The selection of the working frequency range of the system responds to multiple factors: the physical size of the array, the sensor spacing, the frequency response of the acoustic MEMS sensors and the emitter, the acoustic reflectivity of people, and finally the required angular resolution of the system. As a result of all these considerations, it is necessary to reach a compromise relationship, since many of these parameters are opposite. Thus, a frequency band between 14 and 21 kHz has been selected. These high frequency values have been selected also to avoid the ambient noises, of much lower frequency, that could interfere with the behavior of the system. Once the working frequency range was defined, the acoustic array was designed in such a way that the sensor spacing provided good resolution at low frequencies, and it also avoided the appearance of grating lobes in the Field of View (FoV) for high frequencies.

SPH0641LU4H-1 digital MEMS microphones of Knowles were chosen for the implementation of the array. These microphones have a PDM (Pulse Density Modulation) interface and a one-bit digital output [49]. The main features of these microphones are: high performance, low-power, omnidirectional response, 64.3 dB SNR, high sensitivity (−25 dBFS), and an almost flat frequency response (±2 dB in the range of 10 kHz to 24 kHz).

#### 2.2.2. Processing System

The hardware used to implement the system was selected taking into account the previous requirements defined in Section 2.1. A search for a commercial solution was done due to the high cost and time of a specific hardware design.

The base unit of the system is an sbRIO 9629 platform [50]. This platform belongs to National Instruments, particularly to the Reconfigurable Input-Output (RIO) family of devices. Specifically, this sbRIO platform is an embedded single-board controller, with an FPGA Artix-7 200T and a Quad-Core Intel Atom processor. The FPGA has 96 digital inputs/outputs, of which 75 are used as the connection interface with the 150 MEMS microphones of the array, so that in each I/O line, two microphones are multiplexed, while the other lines are used to generate the clock and synchronize. The Atom processor is equipped with 2 GB of DDR3 RAM, 4 GB of built-in storage space, USB Host port, and Giga Ethernet port. Finally, it has sixteen 16-bit analog inputs and four 16-bit analog outputs, which are used to generate the transmitted signal. All this hardware is mounted on a single PCB board (155 mm × 102 mm × 35 mm) with a weight of 330 g. These sbRIO devices are oriented to sensors with nonstandard acquisition procedures, allowing low-level programming of the acquisition routines.

The embedded processor included in sbRIO is capable of running all the software algorithms to detect targets, so it can be used as a standalone array module formed by a sbRIO connected to a MEMS array board as shown in Figure 2. The position of the detected target will be sent to the AEB system of a vehicle.

### 2.3. Software Algorithms

Based on being active, this system must consist of both a receiving subsystem, based on an acoustic array, and a transmitting subsystem, in charge of generating the signal(s) that will be reflected on the possible pedestrians to be detected. Thus, the algorithms implemented in the system, shown in Figure 3, can be divided into four blocks: Transmission Generation, MEMS Acquisition, Signal Processing, and Detection.

The Transmission Generation block synthesizes a pulsed multitone signal to be sent through the DA converter to the signal amplifier and from there to the loudspeaker which will output the transmitted signal.

In the Acquisition block, implemented in the FPGA, each MEMS microphone acquired the reflected signal with a PDM interface. This interface internally incorporates a one-bit sigma-delta converter with a sampling frequency of 2 MHz. In this block, a common clock signal is generated for all 150 MEMS to read signals simultaneously via the digital inputs of the FPGA.

In the Signal Processing block, three routines are implemented:Delay and Sum beamforming: a discrete set of beams that cover the surveillance space in azimuth is generated for a fixed elevation angle. Each of these beams is implemented using a Delay and Sum beamformer, which essentially delays specifically each of the 150 microphone signals and sums them together, so that the array points in a specific direction in the surveillance space. A diagram of a beamformer can be observed in Figure 4a.Decimation and Filtering: applying downsampling techniques, based on decimation and filtering [51], 150 independent signals are obtained and the sampling frequency is reduced from 2 MHz to 50 kHz.Matched filter: A matched filter is then applied to the decimated signal to maximize the SNR at the input of the detection block.

The Detection block implements three processes:First, the relative maxima for each of the beams are identified and a list of potential targets is generated.Then, all targets that are outside the detection lane are eliminated, while those whose distance is within the surveillance range of the system [Rmin, Rmax] are selected, as shown in Figure 4b.Finally, the selected targets are processed by a CFAR (Constant False Alarm Rate) detector [52]. The CFAR detector compares the energy of the potential target with a dynamic threshold that is proportional to the average of the energy received in a set of cells close to the evaluated one, where a number of cells contiguous to the evaluated one, called guard cells, have been excluded. Its scheme can be observed in Figure 4c. The CFAR threshold has been obtained by varying the CFAR gain according to the equation: −*n*(1)$$Threshold\left[n\right]=k\cdot \frac{1}{N}\left\{\overset{n-n1}{\underset{m=n-n2}{\sum }}x\left[m\right]+\overset{n+n2}{\underset{m=n+n1}{\sum }}x\left[m\right]\right\}$$
where *n*1 and *n*2 determine the indices of the cells where the threshold is evaluated. Specifically, the threshold is evaluated in 2 × (*n*2 − *n*1) cells that are n1 cells away from the cell under test. On the other hand, *k* is the parameter that allows weighting the relationship between the Detection Probability and the False Alarm Probability. The values of *n*1 and *n*2 are calculated as a function of the transmitted pulse width.

## 3. Results

### 3.1. Test Setup

For the analysis of the system, a compact vehicle has been assumed on a normal road surface and an ABS braking system with a maximum deceleration of 0.8 g ms^−2^. A 4 m wide road has been used, with street lamps and trees along its edges, as shown in Figure 5a.

Taking into account that the typical vehicle speed limits in urban environments are between 30 km/h and 50 km/h, the minimum braking distance of a vehicle travelling at these speeds was calculated, as shown in Table 1. On the basis of these braking distances, a set of experiments has been designed, placing a person at six distances from the system: 5 m, 7.5 m, 10 m, 12.5 m, 15 m, and 20 m, defining the six detection distances to be considered on the analysis of the system.

It can be observed that for the speed of 30 km/h, all the defined test distances would be valid, i.e., if a pedestrian is detected at these distances, the AEB system would be able to brake the vehicle so that no collision would occur. For the speed of 40 km/h, only the defined test distances of 10 m and above would be adequate to avoid a collision between the vehicle and the pedestrian. For the speed of 50 km/h, only test distances defined from 15 m would be adequate. In the cases for detection distances shorter than the braking distance, it has been confirmed by simulations with CarSim [53], that although the vehicle would collide with the pedestrian, the collision would occur at a sufficiently low speed so that the impact would not cause serious injuries to the pedestrian.

The time required to detect a pedestrian is the time required for the transmitted signal to propagate to the pedestrian and be reflected back to the sensor. In addition, the time needed for signal processing (beamformer, filters, and detector) has to be added. The scenario configuration has a maximum range of 25 m, which, assuming a propagation speed of 343 m/s, takes 145 ms for the acoustic signal to travel. Signal processing is performed on the FPGA in real time simultaneously with the acquisition. At the end of the acquisition, the processed signal is already formed for all 11 beams. Additionally, 25 ms are required for the detection process to be implemented in the embedded multicore processor. Therefore, the system is capable of performing five detections per second, as the total time required for each detection is less than 200 ms.

For the tests carried out, a transmission signal was generated consisting of a 3 ms pulse composed of eight discrete frequencies between 14 kHz and 21 kHz, with a spacing of 1 kHz. Several frequencies have been used to improve the probability of detection, since depending on the physical characteristics and clothing of pedestrians, reflectivity varies with frequency [54]. Frequencies below 14 kHz have not been used because this would result in beam broadening, which would reduce the spatial resolution of the system. The pulse width of 3 ms was chosen as a compromise between the range resolution, which is inversely proportional to the pulse width, and the transmitted energy. The signal was generated with a tweeter loudspeaker with a flat frequency response (±1.5 dB) between 10 kHz and 24 kHz.

### 3.2. Scenario Analysis

For a person located at 10 m (Figure 5a), an acoustic image was obtained at the operating frequency of 20 kHz, generating 120 beams of 4° of beamwidth with steering azimuth angles from −60° to 60°, with 1° resolution and for a surveillance range between 4 and 25 m. In this case, Figure 5b represents the obtained acoustic image in azimuth/range space where the red crosses represent the detected targets. In the image, it is clear that the system detects both the person, placed at around 10 m, and other objects in front of it, such as lampposts and trees on the roadside. Table 2 shows the targets detected, with their position, expressed in range and azimuth. Comparing Figure 5a,b, it can be observed that in the acoustic image (Figure 5b) two detections appear that are not represented in the test scenario (Figure 5a). These targets actually exist in the scenario but do not appear in Figure 5a because the picture is not taken with a wide-angle camera.

In Figure 5b, it can also be visualized the lane boundaries, represented by a dashed green line. As shown in Figure 5b, most of the detected targets are outside the road lane and should therefore not be taken into consideration by our system. It is therefore justified to perform a selection of the targets that are inside the road in the preset surveillance range, from 5 to 25 m. This operation is performed by the detection algorithm (Figure 4c). After applying the lane filter, only the pedestrian remains as a detected target.

### 3.3. Detector Performance Analysis

In a real system, it is not necessary to know with such a high resolution the position of a pedestrian and on the other hand fast update rates between 5 and 10 frames per second are required, and it is advisable to work with a smaller number of beams.

First, the angles of excursion for the detection system must be determined. Assuming that the road has a width of *d* meters and that the car drives in the center, at a distance *R*, the angle of maximum excursion will be a function of distance *R* according to the equation:(2)$$\theta max=atan2\left(\frac{d/2}{R}\right)$$

In turn, the array used has a beamwidth (Δ*θ*) of 4° at 3 dB for 20 kHz; therefore, the number of beams required is:(3)$$Nbeams=floor\left\{\frac{2\cdot \theta max}{\Delta \theta }\right\}$$

Combining Equations (2) and (3) gives the number of beams required for the distances defined in the test scenario. So, since the system has to detect any pedestrian in the range between 5 and 20 m, the minimum value defined for the distance R shall be used, giving a number of working beams of 11. In this case, the steering angles of the defined beams are equispaced between −20° and 20°, with a separation of 4°. Figure 6 shows the radiation patterns for the 11 defined beams, where it can be observed that they cut off at 3 dB and cover the scan angles between −22° and 22°.

The analysis of the CFAR detector has been carried out as a function of the associated threshold gain k. This value has varied between 3 and 10, with increments of 0.01. Several experiments have been carried out to obtain the values of the guard and reference window cells of this CFAR detector, *n*1 and *n*2, respectively. The values of these cells’ range have been determined based on the pulse width used. In this case, given that the signal pulse used lasts 3 ms, which corresponds to a value in range of 100 cm, both the guard cells and the reference window cells were calculated for an initial 100 cm value. After the study carried out to find suitable values for these cells, a guard cell value of 200 cm and a reference cell value of 300 cm have been selected.

Once the operating parameters of the different algorithms to be used in the system have been defined, 1000 experiments were carried out for each of the six defined test scenarios, placing the pedestrian on the center of the lane. Figure 7 below shows the overall Detection Probability (Pd) and False Alarm Probability (Pfa) obtained for the system. In Figure 8, the Detection Probability (Figure 8a) and the False Alarm Probability (Figure 8b) are displayed in detail for each scenario, as a function of the *k* value that controls the threshold of the CFAR detector.

Figure 7 shows that both the Detection Probability and the False Alarm Probability are monotonically decreasing functions with k. Based on RADAR theory, according to the Neyman-Pearson Observer theorem [52], the optimal detection threshold, and hence the value of the Detection Probability of the system, is obtained from a preset False Alarm Probability value. In this case, a False Alarm Probability value of 10^−2^ has been set, as this is a typical value used for in-vehicle detection systems. Based on this value of the False Alarm Probability, the threshold value obtained corresponds to *k* = 4.91, which in turn defines a value for the Detection Probability of 0.995. For this type of application, where a person’s life is at risk, the obtained Detection Probability is adequate. Furthermore, it should be taken into account that in these systems a False Alarm confirmation procedure is usually established, which allows the false detections obtained to be discarded, and therefore the value of the False Alarm Probability defined, which may seem high, is adequate in practice.

In addition to the overall behavior of the system, the Detection Probability and the False Alarm Probability have also been analyzed for each of the scenarios defined, shown in Figure 8. Analyzing the behavior of the Detection Probability as a function of distance (Figure 8a), it can be seen that for distances of 5 and 7.5 m, values above the Global Detection Probability are obtained (Figure 7); for distances of 15 and 20 m, the Detection Probability values obtained are around the global value; and finally, for distances of 10 and 12.5 m, their Detection Probability values are below the global value. In principle, this probability should decrease as the distance increases, because the SNR decreases. However, this behavior is not observed in the system detector, due to the existence of reflections from the multiple targets located at the road boundaries, which interfere with the reflections from the pedestrian itself.

Figure 8b shows that False Alarm Probabilities are different depending on the distance of the pedestrian when the detection threshold is low. These false detections are due to the sidelobes of the array beampattern, through which reflections from objects close to the lane are received. Depending on the relative position between the pedestrian and the nearby objects, destructive/constructive interference arises and influences the average energy estimation estimated by the CFAR in the detection environment. By increasing the detection threshold, this uneven behavior disappears as the CFAR detector eliminates these false detections. In this work, the experiments have been evaluated independently, and it has not been taken into account that in a dynamic environment, the detections of each experiment can be validated in subsequent experiments by confirming them as detections or discarding them as False Alarms, reducing the False Alarm Probability significantly.

With the value of the threshold obtained (*k* = 4.91), the values of the Probability of Detection and the Probability of False Alarm for each of the scenarios have been calculated individually and are shown in Table 3. In view of these, it can be seen that, for certain scenarios, the value of the Probability of False Alarm exceeds the established limit value of 10^−2^, although for other scenarios, it is lower. To ensure that the limit value for the Probability of False Alarm is met in all scenarios, it has been calculated that the value of k associated with the CFAR threshold to be considered should be 5.07. For this new value of k, it can be seen in Table 3 that all the values of the False Alarm Probability are below it. At the same time, it can also be observed that the average value of the Probability of Detection has decreased by only 0.3%, while the average value of the Probability of False Alarm has decreased by 50%.

In summary, a characterization of the detector has been made, both jointly for the six test scenarios and for each one independently, defining two possible detection thresholds: the first one based on the joint characterization and the second one based on considering the worst case of all the scenarios.

## 4. Conclusions

In this work, an active acoustic system has been presented that, by means of a 2D array of MEMS microphones and a processing architecture based on an FPGA/microprocessor, can estimate the 2D position of a person up to a distance of 20 m and prevent a car from colliding with a pedestrian up to speeds of 50 km/h. For higher speeds, the vehicle’s AEB system, which will interact with the presented system, can mitigate the impact with the pedestrian, avoiding fatal injuries. By estimating the 2D position of the pedestrian, the system could even warn the vehicle’s steering system to avoid the pedestrian.

The analysis of the system for a set of scenarios with people at different distances has shown the feasibility of the system. A processing scheme based on beamforming, range and lane filters, and finally a powerful CFAR detection algorithm has been developed, where the operating parameters of a False Alarm Probability lower than 0.01 and a Probability of Detection higher than 0.99, acquired by more than 6000 experiments, have been obtained.

This system can be fused with existing camera, LIDAR, and RADAR based pedestrian detection systems, decreasing the False Alarm Probability and increasing the joint Detection Probability. Finally, this system can work in low visibility situations: rain, fog, smoke, night, etc., where cameras/LIDAR are not operational; as well as improve the detection of a person by a RADAR system, since the radar section of people is usually very low compared to their acoustic section.

## Figures and Tables

**Figure 1 sensors-21-04162-f001:**
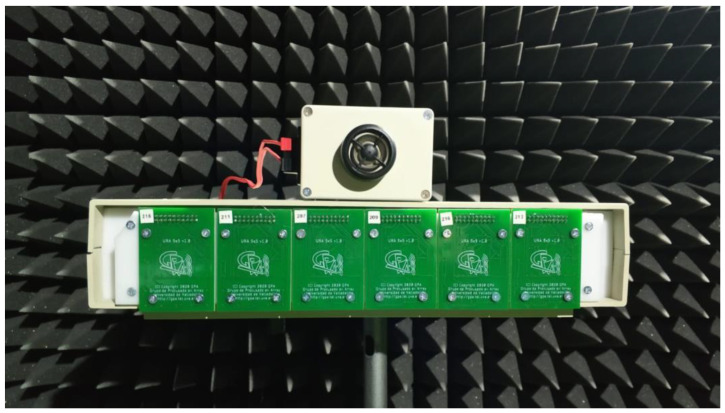
System MEMS array.

**Figure 2 sensors-21-04162-f002:**
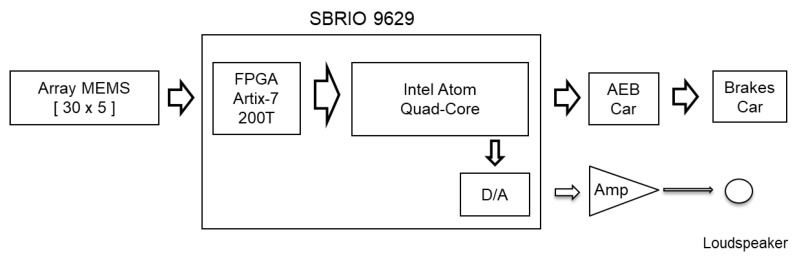
Hardware setup diagram.

**Figure 3 sensors-21-04162-f003:**
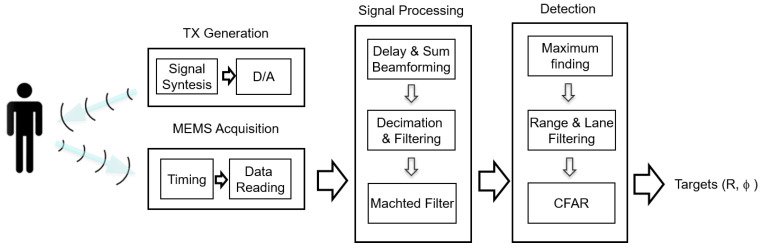
Processing algorithms.

**Figure 4 sensors-21-04162-f004:**
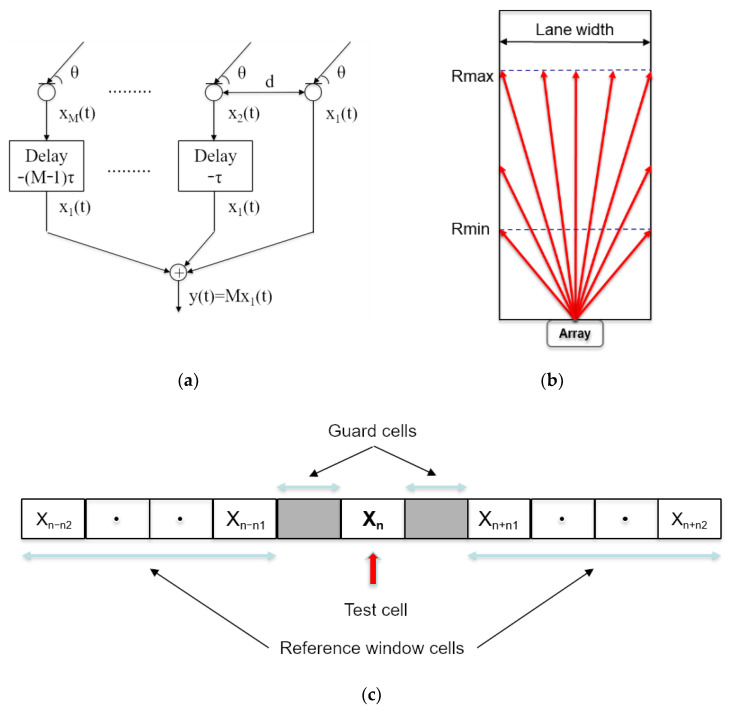
(**a**) Delay and Sum beamformer. (**b**) Lane window. (**c**) CFAR detector.

**Figure 5 sensors-21-04162-f005:**
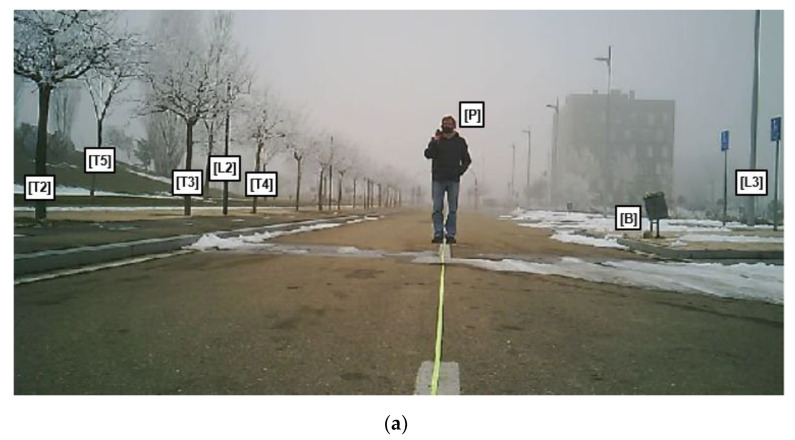
(**a**) Test scenario with a person placed at 10 m. (**b**) Acoustic image with detected targets.

**Figure 6 sensors-21-04162-f006:**
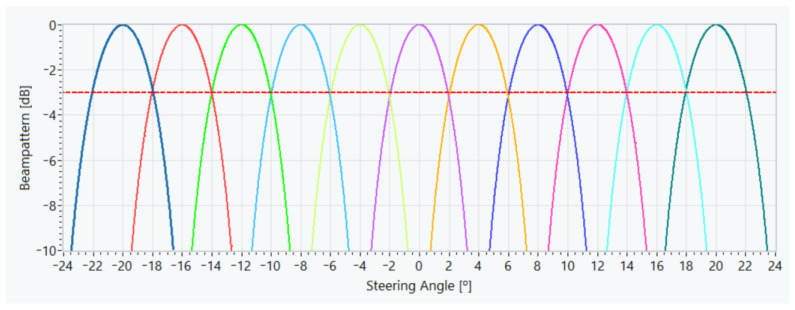
Beampatterns of the 11 defined beams.

**Figure 7 sensors-21-04162-f007:**
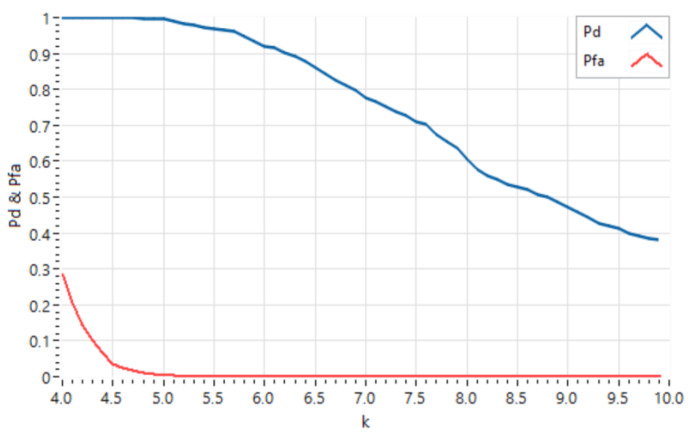
Global Detection Probability and False Alarm Probability of the system.

**Figure 8 sensors-21-04162-f008:**
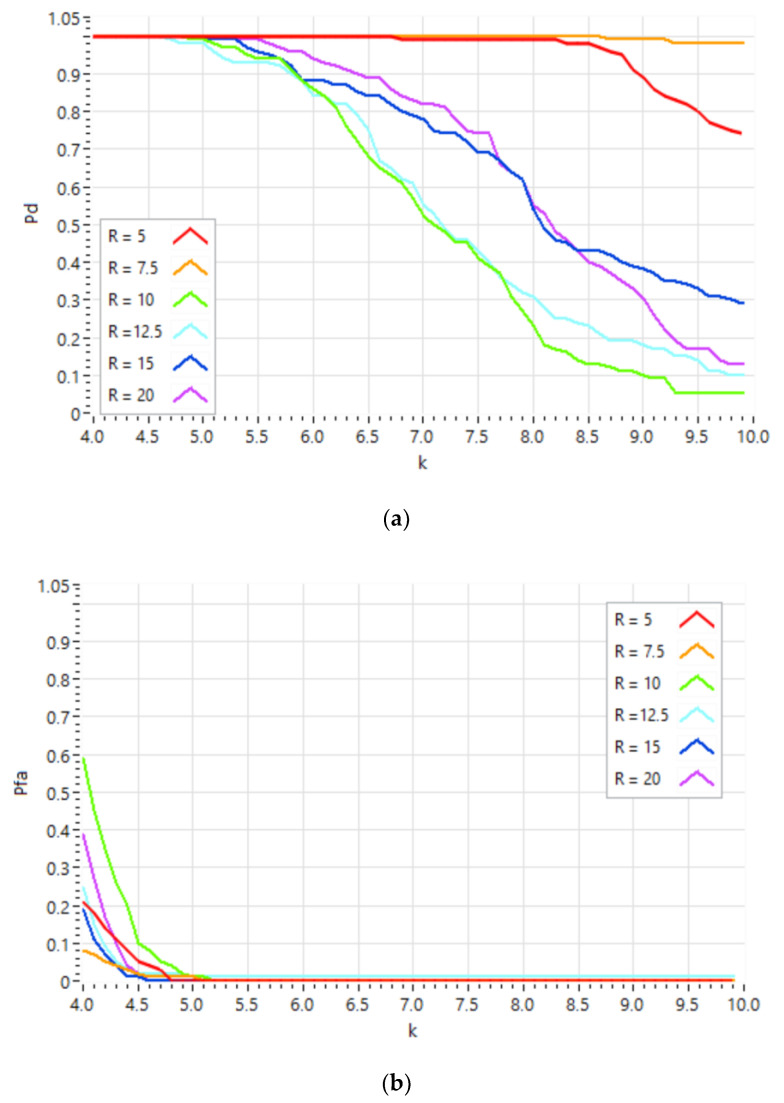
(**a**) Detection Probability and (**b**) False Alarm Probability, for the test scenarios.

**Table 1 sensors-21-04162-t001:** Braking distances for the considered vehicle speeds, with a 0.8 g deceleration.

Speed [km/h]	Braking Distance [m]
30	4.4
40	7.9
50	12.3

**Table 2 sensors-21-04162-t002:** Position (range and azimuth) of the detected targets.

ID	Target	Range [m]	Azimuth [°]
T1	Tree	7.8	−51
P	Pedestrian	10.5	3
T2	Tree	12.6	−28
L1	Lamppost	12.8	42
B	Bin	14.4	19
T3	Tree	18.3	−18
L2	Lamppost	21.3	−15
L3	Lamppost	22.7	26
T4	Tree	24.3	−13
T5	Tree	24.4	−25

**Table 3 sensors-21-04162-t003:** Detection Probability and False Alarm Probability for the test scenarios, for *k* = 4.91 and *k* = 5.07.

R [m]	*k* = 4.91		*k* = 5.07	
	Pd	Pfa	Pd	Pfa
5	0.999	0.003	0.999	0.002
7.5	0.998	0.027	0.997	0.010
10	0.993	0.018	0.992	0.009
2.5	0.984	0.009	0.967	0.007
15	0.997	0.002	0.997	0.001
20	0.999	0.001	0.998	0.001
Average value	0.995	0.010	0.992	0.005

## Data Availability

The data used in this work may be requested by sending an e-mail to one of the authors of the article.

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
