# Peer review of "Feasibility of Using a MEMS Microphone Array for Pedestrian Detection in an Autonomous Emergency Braking System"

_sensors, 2021, doi:10.3390/s21124162_

Round 1

Reviewer 1 Report

This paper suggests an active sound system has been developed, which with a sensor based on a 2D array of MEMS microphones, working in the 14kHz to 21kHz band, allows the detection and estimation of the position of people. The Paper is interesting and well written. Below are my comments and recommendations.

  • The abstract is too short. Please have about 200 words in the abstract. Discuss the major findings of the paper in the abstract. Also briefly discuss the importance of the project.
  • What is the error in the system? When the car turns or get close to an obstacle such as a tree or wall (while parking) would the sensor gives false alarms? What are the considerations to remove the false alarm?
  • Please state the model and the company that provided MEMS microphones.
  • State why the acoustic frequency range between 14 and 21kHz is selected?
  • Authors need to combine the figures as much as possible to make the paper easier to read.
  • How long does it take for the algorithm to detect an obstacle effectively?
  • Please discuss how the quality and accuracy of the system can be improved for the future work

Author Response

First of all, thank your for your comments.

Here are the responses to each of them:

  • The abstract is too short. Please have about 200 words in the abstract. Discuss the major findings of the paper in the abstract. Also briefly discuss the importance of the project.

As you can observe, on the abstract of the new manuscript, we have extended it as you suggested us:

Abstract: Pedestrian detection by a car is typically performed using camera, LIDAR or RADAR-based systems. The first two systems, based on the propagation of light, do not work in foggy or poor visibility environments, and the latter are expensive, and the probability associated with their ability to detect people is low. It is necessary to develop systems that are not based on light propagation, with reduced cost and with a high detection probability for pedestrians. This work presents a new sensor that satisfies these three requirements. An active sound system, with a sensor based on a 2D array of MEMS microphones, working in the 14 kHz to 21 kHz band, has been developed. The architecture of the system is based on a FPGA and a multicore processor that allow the system to operate in real time. The algorithms developed are based on a beamformer, range and lane filters, and a CFAR (Constant False Alarm Rate) detector. In this work, tests have been carried out with different people and in different ranges, calculating in each case and globally the Detection Probability and the False Alarm Probability. The results obtained verify that the developed system allows the detection and estimation of the position of pedestrians, ensuring that a vehicle travelling at up to 50 km/h can stop and avoid a collision.

  • What is the error in the system? When the car turns or get close to an obstacle such as a tree or wall (while parking) would the sensor gives false alarms? What are the considerations to remove the false alarm?

This system is not intended as a parking assistant, for which commercial systems based on short-range ultrasonic sensors already exist. The developed sensor would be used in AEB (Autonomous Emerging Braking) systems. In this case, when there is a detection, it is verified by a new experiment, confirming the detection, in which case the AEB system would act, or discarding it, if it has been a false alarm. You can see this explanation in the paragraph of the introduction, between lines 75 and 82:

“Under these premises, the idea of analyzing the feasibility of detecting pedestrians by means of an acoustic array installed in a vehicle arose. The system, working in conjunction with AEB systems, would prevent traffic accidents. Using acoustic signals ensures that the system will function properly in environments with reduced visibility. Arrays are ordered sets of identical sensors, whose response pattern is controlled by modifying the amplitude and phase given to each sensor, their spacing and their distribution in space (linear, planar, etc.) [39]. Beamforming techniques are used with arrays [40], so that the array pattern is electronically steered to different spatial positions.”

It is true that for future developments, the system could be adapted to detect pedestrians in the parking maneuver.

  • Please state the model and the company that provided MEMS microphones.

This information has been already explained in section 2.2.1, where the MEMS microphone array is described. As you can observe in lines 160-164:

“SPH0641LU4H-1 digital MEMS microphones of Knowles were chosen for the imple-mentation of the array. These microphones have a PDM (Pulse Density Modulation) in-terface and a one-bit digital output [49]. The main features of these microphones are: high performance, low-power, omnidirectional response, 64.3 dB SNR, high sensitivity (−25 dBFS) and an almost flat frequency response (±2 dB in the range of 10kHz to 24kHz).”

  • State why the acoustic frequency range between 14 and 21kHz is selected?

The second paragraph of section 2.2.1 (lines 149-159) has been modified to clarify the justification of the frequency range selection:

“The selection of the working frequency range of the system responds to multiple factors: the physical size of the array, the sensor spacing, the frequency response of the acoustic MEMS sensors and the emitter, the acoustic reflectivity of people and finally the required angular resolution of the system. As a result of all these considerations, it is necessary to reach a compromise relationship, since many of these parameters are opposite. Thus, a frequency band between 14 and 21 kHz has been selected. These high frequency values have been selected also to avoid that ambient noises, of much lower frequency, could interfere with the behavior of the system. Once the working frequency range was defined, the acoustic array was designed in such a way that the sensor spacing provides good resolution at low frequencies, and it also avoids the appearance of grating lobes in the Field of View for high frequencies.”

  • Authors need to combine the figures as much as possible to make the paper easier to read.

Figures 1 and 4 have been combined as the new Figure 3. We have also combined figures 5, 6 and 7 in the new Figure 4. And finally, we have eliminated figure 8.

  • How long does it take for the algorithm to detect an obstacle effectively?

Indeed, this data is very important and we had not included it in the paper, thank you very much for this comment. This information has been included in section 3.1 (lines 261-270).

The time required to detect a pedestrian is the time required for the transmitted signal to propagate to the pedestrian and be reflected back to the sensor. In addition, the time needed for signal processing (beamformer, filters and detector) has to be added.

The scenario configuration has a maximum range of 25 meters, which, assuming a propagation speed of 343 m/s, takes 145 ms for the acoustic signal to travel. Signal processing is performed on the FPGA in real time simultaneously with the acquisition. At the end of the acquisition, the processed signal is already formed for all 11 beams. Additionally, 25 ms are required for the detection process to be implemented in the embedded multicore processor.  Therefore, the system is capable of performing 5 detections per second, as the total time required for each detection is less than 200 ms.

  • Please discuss how the quality and accuracy of the system can be improved for the future work

The quality of the system, meaning the probability of detection and false alarm, based on classical radar detection theory, is improved by increasing the bandwidth and power of the transmitted signal. In our case, the bandwidth could be increased by increasing the number of frequencies, but it would mean an increase of the computational load and in relation to the transmitted acoustic power, if it were high, it could be dangerous for the health and the environment.

We hope to clarify your doubts and correct the mistakes.

Yours sincerely.

The authors

Reviewer 2 Report

This is an interesting manuscript dealing with an important problem, and in particular dealing with a fairly original (possibly innovative) strategy for Autonomous Emergency Braking system based on an array of acoustic sensors (MEMS microphones). The manuscript presents the hardware and software, but the emphasis is on the data processing part, and in particular the experimental validation.

Overall I enjoyed reading this manuscript, however there are a number issues that have to be addressed by the authors before this manuscript can be accepted. Let me start with some major concerns:

  • My biggest concern with this work, is related to the actual choice of the hardware and sensors. Why using an acoustic signal? This is too brievely motivated by authors in the Introduction. We all know the limitations of some on-board sensors mounted on autonomous vehicles, in particular those associated with hazardous weather conditions (heavy rain falls, snow storms, etc.). However, I find that the arguments and elements provided by the authors are not sufficient. Active acoustic SODAR is known to suffer from low SNR, etc. Is LiDaR really that inneffective? How about RADAR? I am just not knowledgeable enough about this, and I am therefore unconvinced that a SODAR solution is optimal.
  • My second biggest concern has to do with the level of English and the quality of preparation of the manuscript. The authors should find a way to get their manuscript reviewed by a native speaker (e.g. "the fatal number" instead of "the number of fatalities"). In addition, American English and British English are mixed up. On top of all this, there are numerous typos left behind by the authors: e.g. Line 83: "3In this"

I also have a number of minor concerns listed below:

  • How is noise pollution in an urban environment affecting the performance of this sensory system?
  • In the introduction, the authors may refer to the following recent review article published by MDPI Sensors: Manivannan, Ajaykumar, et al. "On the Challenges and Potential of Using Barometric Sensors to Track Human Activity." Sensors 20.23 (2020): 6786. The part on tracking human activity is highly relevant to pedestrian detection.
  • Line 169: it should be "4 GB" instead of "4G"
  • It is worth noting that units should be separated by a blank space from their associated value: e.g. 50 km/h and not 50km/h.
  • Figure 4 should be better explained.
  • Figure 12: the false alarm probability is extremely important, and the authors should elaborate a bit more on this.

Author Response

First of all, thank you for your comments.

Here are the responses to each of them:

  • My biggest concern with this work, is related to the actual choice of the hardware and sensors. Why using an acoustic signal? This is too brievely motivated by authors in the Introduction. We all know the limitations of some on-board sensors mounted on autonomous vehicles, in particular those associated with hazardous weather conditions (heavy rain falls, snow storms, etc.). However, I find that the arguments and elements provided by the authors are not sufficient. Active acoustic SODAR is known to suffer from low SNR, etc. Is LiDaR really that inneffective? How about RADAR? I am just not knowledgeable enough about this, and I am therefore unconvinced that a SODAR solution is optimal.

As to why use an acoustic signal, in essence acoustic systems allow working in environments where light-based systems are not operational and for objects that have a low reflectivity for radar systems. In situations where foggy conditions are frequent, an acoustic wave based system is essential.

Regarding the issue of whether acoustic signals have a low SNR associated with them, when working with an array of 150 microphones and shaping techniques the SNR improves by a factor of 150 (21.7dB) and therefore our system has a high SNR.

As for the comment on whether the SODAR solution is optimal, in the article we do not suggest that the acoustic sensor presented is optimal, but that it can be an additional sensor in a vehicle, where, depending on the scenario, it merges with the information from the rest of the sensors. For each type of situation, information from one or more sensors can be used.

  • My second biggest concern has to do with the level of English and the quality of preparation of the manuscript. The authors should find a way to get their manuscript reviewed by a native speaker (e.g. "the fatal number" instead of "the number of fatalities"). In addition, American English and British English are mixed up. On top of all this, there are numerous typos left behind by the authors: e.g. Line 83: "3In this"

We have made the suggested changes and have also contracted the mdpi proofreading service, which will check the English of the article if it is accepted.

  • How is noise pollution in an urban environment affecting the performance of this sensory system?

En base a la selección de la banda de frecuencias, los ruidos urbanos no afectan al sistema, ya que suelen encontrarse en frecuencia bajas. Las pruebas han sido realizadas en entornos reales donde estaban presentes todo tipo de ruidos ambientales y los resultados obtenidos ponen de manifiesto que no han influido en el comportamiento del sistema.

Based on the selection of the frequency band, urban noise does not affect the system, as it is usually found at low frequencies. The tests have been carried out in real environments where all types of environmental noise were present and the results obtained show that they have not influenced the behavior of the system.

We have included this fact on section 2.2.1 (lines 154-156).

  • In the introduction, the authors may refer to the following recent review article published by MDPI Sensors: Manivannan, Ajaykumar, et al. "On the Challenges and Potential of Using Barometric Sensors to Track Human Activity. "Sensors23 (2020): 6786. The part on tracking human activity is highly relevant to pedestrian detection.

Thank you for pointing out this reference, but we would be grateful if you could indicate where in the text we should reference it, because after reading the publication, we do not see its direct application to the subject of the paper.

  • Line 169: it should be "4 GB" instead of "4G"

You are right, it has been corrected.

  • It is worth noting that units should be separated by a blank space from their associated value: e.g. 50 km/h and not 50km/h.

You are right, it has been corrected.

  • Figure 4 should be better explained.

To make this figure clearer, we have merged it with figure 1, and we have also improved the first paragraph of section 2.3 (lines 190-194):

“Based on being active, this system must consist of both a receiving subsystem, based on an acoustic array, and a transmitting subsystem, in charge of generating the signal(s) that will be reflected on the possible pedestrians to be detected. Thus, the algorithms implemented in the system, shown in Figure 3, can be divided into four blocks: Transmission Generation, MEMS Acquisition, Signal Processing, and Detection.”

  • Figure 12: the false alarm probability is extremely important, and the authors should elaborate a bit more on this.

New Figure 8b (Pfa vs. k) has been changed to a zoom of the previous Figure 12b, in order to visualize the behavior of the different Pfas according to k threshold more easily. And also, according to your indications, the justification of Pfa behavior has been expanded in the paper (lines 371-382):

“Figure 8b shows that False Alarm probabilities are different depending on the distance of the pedestrian when the detection threshold is low. These false detections are due to the sidelobes of the array beampattern, through which reflections from ob-jects close to the lane are received. Depending on the relative position between the pedestrian and the nearby objects, destructive/constructive interference arises and in-fluences the average energy estimation estimated by the CFAR in the detection envi-ronment. By increasing the detection threshold, this uneven behavior disappears as the CFAR detector eliminates these false detections. In this work, the experiments have been evaluated independently, and it has not been taken into account that in a dynamic environment, the detections of each experiment can be validated in subse-quent experiments by confirming them as detections or discarding them as false alarms, reducing the False Alarm Probability significantly.”

We hope to have answered all your doubts and corrected the mistakes.

Yours sincerely.

The authors

Round 2

Reviewer 2 Report

The authors did a good job at addressing my comments. I support publication at this stage.